# Low temperature deposition of functional thin films on insulating substrates enabled by selective ion acceleration using synchronized floating potential HiPIMS

Jyotish Patidar, Oleksandr Pshyk, Kerstin Thorwarth, Lars Sommerhäuser & Sebastian Siol ✉

Ionized physical vapor deposition techniques, such as high-power impulse magnetron sputtering (HiPIMS), are gaining popularity but face challenges for deposition on insulating materials, where applying negative potentials for ion acceleration is difficult. While radio frequency biasing works on insulators, it risks film damage from energetic process gas ions. Here, we present Synchronized Floating Potential HiPIMS (SFP-HiPIMS), which exploits the substrate's transient negative floating potential during HiPIMS discharges. By timing the ion arrival with this negative potential, selective metal-ion acceleration can be achieved, improving adatom mobility while minimizing energetic $Ar^+$ bombardment. As proof-of-concept, we deposit $Al_{0.88}Sc_{0.12}N$ thin films on various insulating substrates. SFP-HiPIMS improves the films' crystallinity, texture and residual stress, and also enables epitaxial growth on c-cut sapphire at temperatures as low as 100 °C. SFP-HiPIMS provides a solution for a long-standing challenge in physical vapor deposition, which works for many different materials and integrates readily with standard deposition equipment.

In recent years, ionized physical vapor deposition (IPVD) techniques, such as high-power impulse magnetron sputtering (HiPIMS), have garnered significant attention due to their ability to enhance the quality and performance of thin films in various applications[1,2]. In HiPIMS, power is delivered to the sputter target in short pulses, enabling higher peak power and current densities. The resulting increased plasma density, in turn, facilitates ionization of the sputtered species. Depending on the deposition parameters ionized flux fractions of over 50 % can be achieved[2]. This offers unique opportunities for process design, since ions can be accelerated or deflected using electric and/or magnetic fields. Applying a negative potential to the substrate increases the kinetic energy of ions, which enhances adatom mobility through momentum transfer to the growing film. Consequently, ion bombardment during film growth facilitates the deposition of dense films with improved crystalline properties, even at low

growth temperatures[3,4]. Moreover, ion deflection through substrate biasing has been successfully used for decades in applications such as hard coatings[1] or metallization of vias and trenches in semiconductor device fabrication[5].

While ion bombardment through biasing offers significant advantages, one downside of using a negative bias on the substrate is that it also attracts abundant process gas ions, such as $Ar^+$, from the plasma. These $Ar^+$ ions can be implanted at interstitial sites within the lattice, leading to the formation of point defects and inducing undesirable compressive stress in the system[6,7]. Processes like metal-ion synchronized HiPIMS (MIS-HiPIMS) can mitigate the detrimental effects of $Ar^+$ ion bombardment by selectively accelerating only the film-forming metal ions[8,9]. This method leverages the different time of flight (ToF) of ions following the HiPIMS discharge. During HiPIMS, for each pulse, the process gas ions tend to arrive earlier at the substrate

Empa, Swiss Federal Laboratories for Materials Science and Technology, Dübendorf, Switzerland. ✉ e-mail: Sebastian.Siol@empa.ch

than the corresponding metal ions. This is primarily due to the ionization of working gas within the plasma volume during the onset of the HiPIMS pulse. These working gas ions have a shorter travel distance compared to the sputtered species, which must first be ejected from the target before reaching the substrate[10]. By tailoring the substrate bias potential based on the ToF information, specific ions can be preferentially accelerated while avoiding an unwanted acceleration of the process gas ions. This is instrumental in avoiding defect formation and process gas incorporation into the growing film[8,11]. MIS-HiPIMS has gained a lot of popularity in recent years, as it enables the low temperature deposition of high-quality ceramic thin films with minimal compressive stress and structural defects[8]. Most importantly, it holds the promise to use HiPIMS deposition techniques for more defect-sensitive applications, like the development of optical coatings or even semiconducting thin films. We have recently demonstrated that MIS-HiPIMS can produce high-quality and compact AlN and AlScN films at comparably low deposition temperatures, which exhibit a piezo-electric response comparable to that achieved with commercial state-of-the-art processes[11,12].

Despite the apparent promise of synchronized ion-acceleration during HiPIMS, one long-standing challenge is holding back its potential use for many applications. While synchronized HiPIMS approaches work well for conductive substrates, their application on insulating or electronically floating substrates is limited due to the challenges in applying electric potentials. This also applies to the growth of thick dielectric films, which, depending on their thickness, show a similar behavior as insulating substrates[13]. A process capable of achieving selective ion acceleration on those materials would be valuable for future technologies as the demand for high-quality and low-temperature film deposition on insulating substrates is continually increasing. The forthcoming IoT and AI revolution will necessitate a higher integration, but also deposition on temperature sensitive substrates and device stacks for applications such as microelectromechanical systems (MEMS), energy conversion and storage, robotics or biomedical devices[14–19].

A practical approach to induce a negative charge on an electrically isolated surface is to expose it to a plasma. When an isolated object is immersed in a plasma it receives a higher flux of electrons than ions due to the electrons' higher thermal velocity. As a result, a plasma sheath is formed around the substrate and the surface charges negatively. As the negative potential increases, so do the electron repulsion and ion acceleration until eventually a net zero current is reached. The potential at net zero current is commonly referred to as the floating potential $U_F$. Such floating potentials commonly arise on electrically isolated substrates during plasma-based physical vapor deposition processes. They are an important parameter for such processes as they effectively accelerate ions onto the substrate's surface during deposition.

At present, the most common way to accelerate ions on an insulating substrate involves applying a radio frequency (RF) plasma to the substrate. This process induces a negative self-bias potential on the substrate, driven by the imbalance in electron and ion fluxes, resulting from the higher mobility of electrons compared to the ions[20,21]. Similar to the generation of the floating potential, the substrate will charge negatively until a net zero current is reached. In the case of RF biasing, this potential is referred to as direct current (DC) self-bias potential. RF biasing is common in IPVD on insulating materials. Few studies even demonstrate a synchronization of the RF self-bias with the HiPIMS discharge[22]. However, this approach has limitations for defect-sensitive materials as we demonstrate in this study. Since RF substrate biasing requires a sufficient plasma density in the vicinity of the substrate, an acceleration of process gas ions towards the substrate is hard to avoid. In addition, it is often difficult to maintain a constant DC self-bias, which also affects ion acceleration[22]. This makes it challenging to selectively accelerate only the film forming species, while

simultaneously minimizing the detrimental acceleration of process gas ions. Another commonly reported approach in the literature is bipolar-HiPIMS, where a positive pulse is applied to the target to raise the plasma potential, generating a potential gradient towards the substrate and thus accelerating the sputtered ions[23,24]. While this approach works well for grounded substrates, it is less effective for insulating or electrically floating substrates. For these cases, the substrate's floating potential often rises with the positive pulse preventing an effective acceleration of the ions onto the substrate. Consequently, substantially higher voltages are required to observe any significant effects on the films' growth[13,25].

In this work, we demonstrate an original approach for pulsed magnetron sputtering, Synchronized Floating Potential HiPIMS (SFP-HiPIMS), which uses the substrate's floating potential to accelerate ions onto the growing film. During magnetron sputtering (especially when unbalanced magnetrons are used) the substrate is immersed in the sputter plasma. For electrically isolated substrates this results in the formation of a negative floating potential. For HiPIMS discharges, this floating potential is often more pronounced than for DC processes. In addition, it is mostly constrained to the duration of the HiPIMS pulse and occurs almost immediately when the sputter pulse is applied. Here we demonstrate that this transient negative potential can be used to accelerate positively charged ions onto the substrate. Based on the information of ToF of ionic species, the HiPIMS pulses can be timed in such a way that the floating potential generated by one pulse accelerates the metal ions generated from another pulse. In a proof-of-principle study, we demonstrate the deposition of high-quality, textured AlScN thin films on different insulating or electrically floating substrates at low deposition temperatures. In this study, the Sc-ions originating from one magnetron are accelerated onto the growing film by means of a negative floating potential, which is induced on the substrate by an appropriately synchronized Al discharge from another magnetron. This deposition scheme can be implemented on standard deposition equipment and is broadly applicable to many materials and substrates. It therefore holds the potential to facilitate the sustainable low-temperature deposition of high-quality thin films for many applications and emerging technologies.

## Results

### Synchronized floating potential HiPIMS: SFP-HiPIMS

During sputtering from an unbalanced magnetron, the substrate is exposed to the sputter plasma. If the substrate is electrically isolated, i.e. insulating or not sufficiently grounded, a negative floating potential $U_F$ is formed on its surface[20]. SFP-HiPIMS works on the principle of using the negative floating potential $U_F$ to accelerate specific sputtered ions.

Figure 1A represents the potential distribution in a sputter chamber for a single magnetron. Shown are the cathode (i.e. the sputter target), the electrically floating substrate and an anode. Here, the anode represents all grounded chamber parts. When an electrically isolated substrate is exposed to a sputter plasma, it acquires a negative floating potential ($U_F$) due to the higher mobility of electrons compared to ions, leading to the formation of a plasma sheath. Since the floating potential is typically lower than the plasma potential ($U_P$), the resulting potential difference ($U_P$-$U_F$) accelerates positive ions across the sheath region towards the substrate surface[20]. It is noteworthy that Fig. 1A is a 2-dimensional representation of potential distribution for a single magnetron, while Fig. 1B schematically represents the SFP-HiPIMS approach for two magnetrons.

During HiPIMS, the discharge at the magnetron is pulsed; consequently, the substrate is immersed in the sputter plasma predominantly during the sputter pulses. The HiPIMS discharge is initiated as soon as power is applied to the target followed by a period of rapid plasma expansion. As the plasma density increases during the

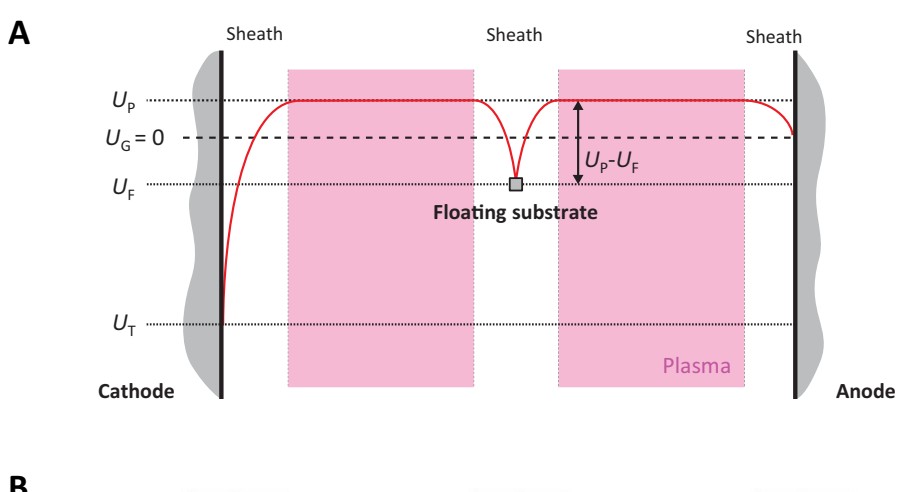

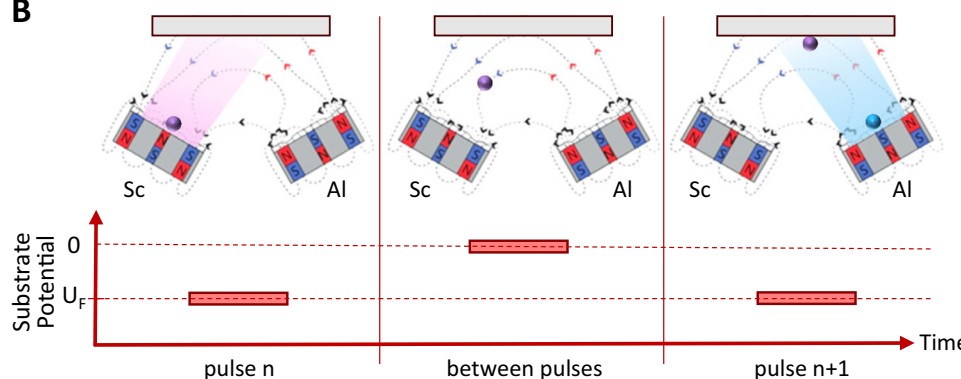

**Fig. 1 | Floating substrate potential utilized to accelerate ions. A** Schematic illustration of a potential distribution in a sputter chamber showing the floating potential on an insulating substrate ($U_F$), the plasma potential ($U_P$) as well as the chamber and target potentials ($U_G$ and $U_T$, respectively). (Adapted from Handbook of Deposition Technologies for Films and Coatings, Third Edition, Walten et. al, Chapter 2 – Plasmas in Deposition Processes, Page 50, 2010, with permission from Elsevier[20]). **B** If two magnetrons are used, the ions generated by a HiPIMS pulse on one magnetron (purple) can be accelerated by the floating substrate potential induced by a later HiPIMS pulse on another magnetron (blue). In between the pulses, the floating potential goes to ground when there is no plasma.

pulse, the electron flux to the substrate also increases, leading to a deepening of the negative floating potential, as apparent in Fig. 2. As the floating potential becomes more negative, both ion bombardment and electron repulsion intensify, eventually leading to a dynamic equilibrium of the floating potential. When the HiPIMS pulse ends, the plasma density decreases, and the floating potential gradually returns to its pre-pulse state. SFP-HiPIMS makes use of this temporally restricted negative floating potential to accelerate specific ions when they arrive at the substrate.

The negative floating potential accelerates all positive ions in the sheath region of the substrate. This also includes the thermalized process gas ions which might still be present during the HiPIMS off-cycle. However, it is possible to time the plasma composition and negative floating potential in a way that leads to the preferential acceleration of certain ionic species. For preferential acceleration of metal-ions, the sputter pulses must be synchronized in a way that a negative floating potential is induced shortly before or in the time frame when the metal-ions are in the direct vicinity of the substrate.

The synchronization condition can be described relative to the start of the main HiPIMS pulse that supplies the film-forming ions which should be accelerated. The goal of SFP-HiPIMS is to accelerate the film-forming ions from this main HiPIMS pulse by the negative floating potential induced by a subsequent HiPIMS pulse. We define the time offset $t_{offset}$ as the time between the start of these two pulses. The time-of-flight (ToF) of the ions is measured from the beginning of the main HiPIMS pulse to the time the ion flux density on the substrate reaches its maximum. It is proposed that for an effective acceleration of the ions, the offset $t_{offset}$ should be chosen so that the temporal overlap of the incident ion flux and the induced substrate floating potential is maximized. In addition, the synchronization can be adjusted to minimize the acceleration of non-film forming species, such as $Ar^+$ or other process gas ions. To this end, especially further acceleration of the energetic process gas ions immediately following the HiPIMS pulse should be avoided[8].

Several pulse configuration setups can be used for the acceleration of ions, including single or multiple magnetrons. In this study, we demonstrate the feasibility of the SFP-HiPIMS approach for the deposition of crystalline AlScN films at low temperatures. For this, we use a confocal setup with multiple magnetrons. In this configuration, we accelerate the Sc ions due to their larger mass through the floating potential generated by Al pulse (see Fig. 1B). Analogous to the deposition from multiple magnetrons, multiple pulses in the form of a pulse-train can be applied to a single magnetron. Each pulse would generate a burst of ions, which can then be accelerated by the floating potential induced by the following pulse. This way, all ions, but the ones originating from the last pulse in the pulse-train can be accelerated. In an even simpler setup of equipment where pulse-train configuration is not possible, the method would also work for a single magnetron where the frequency can be chosen in such a way that the ToF of ions to be accelerated matches with the time between the HiPIMS pulses, in some instances such conditions could be achieved using pulsed DC sputtering.

Since the method relies on ion acceleration via the generated floating potential of the substrate, its primary limitation is the magnitude of this potential. Typically, this bias is in the range of a few tens

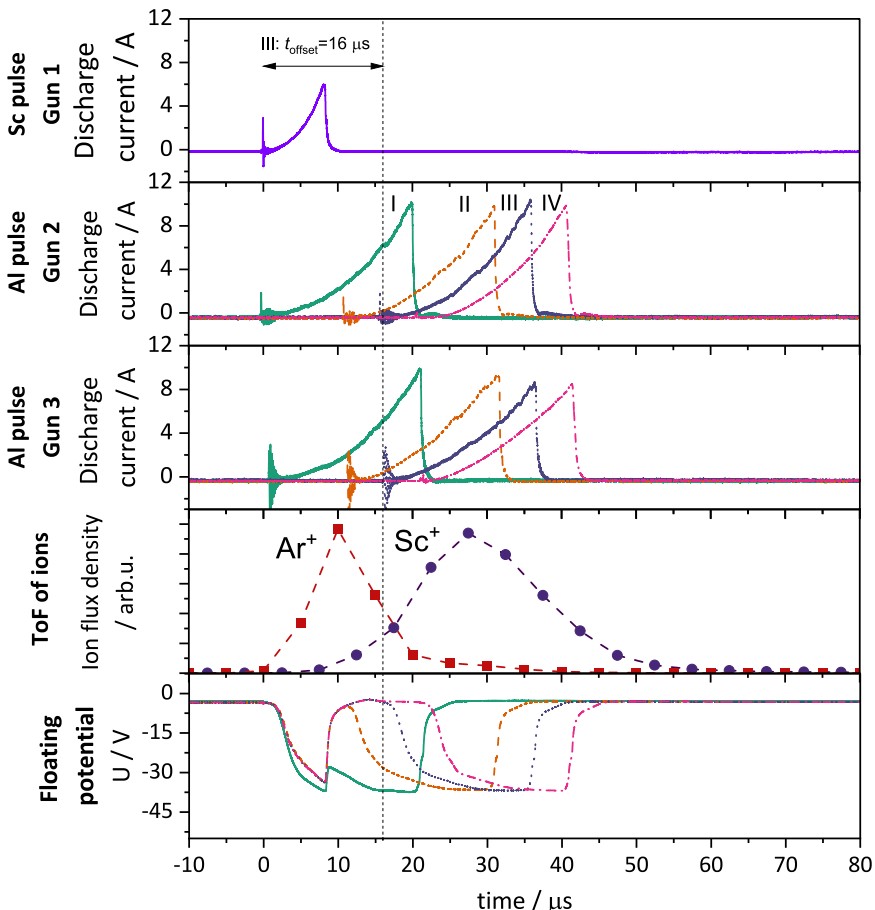

**Fig. 2 | SFP-HiPIMS deposition of AlScN thin films.** The figure shows the discharge currents of the Sc and Al pulses, along with the time-of-flight of $Ar^+$ and $Sc^+$ ions and the substrate floating potential for four different synchronization modes: I: no offset, II: $t_{offset} = 11$ μs, III: $t_{offset} = 16$ μs, IV: $t_{offset} = 21$ μs. The Sc deposition is operated from one magnetron, while the Al deposition is operated from two magnetrons. The dashed lines in the ion flux plot connect the markers and serve as a guide for the eye.

of volts, which restricts the kinetic energy of the ions to less than 50 eV. This energy range is ideally suited to significantly enhance adatom mobility without risking detrimental effects due to high energy ion bombardment. Overall, this approach is most effective for depositing thin-film materials like AlScN with low lattice displacement thresholds and sensitivity towards point defects[26–28]. Further advancements to increase the floating potential (e.g. by means of an electron gun) could significantly broaden its applicability for an even wider range of materials.

## Proof of concept: Low temperature epitaxial growth of $Al_{0.88}Sc_{0.12}N$

We demonstrate the feasibility of the concept of SFP-HiPIMS in a proof-of-principle experiment. In this experiment, Aluminum Scandium Nitride thin films with approximately 12 cation at.% Sc ($Al_{0.88}Sc_{0.12}N$) are deposited on insulating or electrically floating substrates using reactive HiPIMS from two Aluminum (Al) and one Scandium (Sc) magnetrons in reactive $Ar/N_2$ atmosphere in a UHV deposition chamber. AlScN is a piezoelectric and ferroelectric material, which is commonly used for RF-MEMS applications[29–31]. The material has been intensively investigated in recent years due to its potential for RF filters in 5 G and 6 G telecommunication technology[18,32]. We have recently demonstrated that MIS-HiPIMS processes facilitate the growth of high-quality AlScN thin films, even at low substrate temperatures and acceleration bias potentials[11,12]. However, to date, none of these approaches have been demonstrated on insulating or electronically floating substrates.

In the proof-of-principle experiment, all three magnetrons are operated in HiPIMS mode. The pulse patterns for all magnetrons are shown in Fig. 2, along with the respective generated floating potential and ToF of the relevant ions. ToF data for $Ar^+$ and $Al^+$ ions from Al magnetrons are intentionally omitted from the plot to maintain clarity but are available in the supplementary information (see Supplementary Fig. 1). In the usual mode of operation, all discharges are initiated simultaneously (Mode I). For SFP-HiPIMS synchronization, an offset between the Sc and Al pulses is introduced with the goal to accelerate the incident $Sc^+$ ions by means of the substrate's floating potential induced during the Al discharges. Figure 2 shows the discharge currents of the Sc and Al pulses, the measured ToF of $Ar^+$ and $Sc^+$ ions and the substrate's floating potential for four different synchronization modes: I: no offset, II: $t_{offset} = 11$ μs, III: $t_{offset} = 16$ μs, IV: $t_{offset} = 21$ μs. The offset times were chosen to accelerate the film-forming $Sc^+$ ions, but not the $Ar^+$ process gas ions. Herein, we consider III the most promising synchronization scheme, since here the negative floating potential is well aligned with the arrival of $Sc^+$ ions at the substrate. The maximum floating potential reaches around −35 V for the chosen deposition parameters. The magnitude of this potential can be tuned, if necessary, by changing the balancing of the magnetrons, the working distance or other process parameters such as the HiPIMS discharge voltage (see Supplementary Fig. 3).

To benchmark SFP-HiPIMS against the commonly performed RF substrate biasing additional depositions are performed for reference. Typically, achieving a stable RF plasma for substrate biasing requires higher pressures and/or power levels, which poses a significant

challenge for pulsed RF biasing at standard sputtering pressures (3–5 μbar). However, the presence of a HiPIMS plasma facilitates the ignition and stabilization of the RF plasma even at very low powers and pressures, making pulsed RF biasing feasible under these conditions. An overview of the operational boundaries of RF substrate biasing for our deposition tool can be found in Supplementary Fig. 4. Building on this understanding, we investigate samples deposited using a continuous RF biasing (c-RF) as well as a pulsed RF substrate bias synchronized with the Sc$^+$ metal-ion flux (p-RF). In both these modes, the DC self-bias is tuned to be comparable to the floating potential observed during SFP-HiPIMS. The detailed substrate potential profiles for these deposition modes are provided in Supplementary Fig. 5.

A detailed materials characterization shows significant improvements in the crystalline quality and texture of the films in the case of the SFP-HiPIMS deposition modes (II-IV), when compared to the conventional mode of operation (I). Figure 3 shows measurements of the structural properties as well as the residual stress for AlScN films deposited using the four different synchronization modes as well as for the two reference samples deposited using RF substrate biasing. In addition, pole figures are added for textural analysis of the AlScN films grown on sapphire (001) for conventional and SFP-HiPIMS. The results show an increase in crystallinity and texture for the SFP-HiPIMS depositions, which is indicated by the strongly reduced Full Width at Half Maximum (FWHM) of the (002) rocking curve as well as the disappearing of the AlScN (102) peak in the θ−2θ measurements. Without any deliberate ion acceleration, the AlScN films exhibit tensile residual stress. This is a common observation in this material as Sc-doping leads to a distortion of the wurtzite lattice[12,33]. As expected, the energetic Sc$^+$-ion bombardment during SFP-HiPIMS leads to a reduction in tensile residual stress from 850 MPa to an almost stress-free state. Most notably, the SFP-HiPIMS deposition mode enables hetero-epitaxial growth of AlScN on sapphire (001) at remarkably low temperatures of 100 °C, while the deposition without synchronization leads to random in-plane orientation (Fig. 3C).

When comparing these films to those deposited using RF biasing, we observed that while RF biasing led to an improved texture as indicated by the reduced FWHM of the (002) rocking curve, it also resulted in significant compressive stress. The increase in compressive stress is accompanied by Ar$^+$-ion incorporation in the AlScN films. Rutherford backscattering spectroscopy (RBS) analysis of the as-grown films shows Ar$^+$-ion concentrations of approximately 1.2 at.% and 0.8 at.% for the continuous and pulsed RF biases, respectively. In contrast, films without RF substrate biasing exhibit no measurable Ar$^+$-ion incorporation (see Supplementary Figs. 6 and 7). This striking difference can be explained by the undesirable acceleration of process gas ions during RF biasing. As the RF plasma at the substrate is formed mainly from process gas ions, an acceleration of these ions on the substrate is unavoidable. Ion energy measurements performed using a retarding field energy analyzer (RFEA) mounted to the substrate holder indicate that during RF biasing with a DC self-bias of −31 V, the Ar$^+$-ions exhibit kinetic energies of up to 60 eV which exceeds reported lattice displacement energies in wurtzite AlN (see Supplementary Fig. 8)[28,34]. This high value can be explained by the high acceleration voltages during RF biasing and the variations in self-bias over the HiPIMS cycle (see Supplementary Fig. 5). In contrast, during the HiPIMS pulse off-time, the thermalized process gas ions exhibit much lower energies, typically around 1-2 eV. The average ion energy over a complete HiPIMS cycle, measured using RFEA, was approximately 4 eV, in line with the measured average floating potential of −2.5 V. Even if the thermalized process gas ions get accelerated during the HiPIMS pulse, when the floating potential is maximized, their energy would still be significantly lower compared to the RF biasing case. This significant difference in process gas ion energies highlights why RF biasing inherently induces higher compressive stress in the deposited films.

The positive effects of the SFP-HiPIMS mode are further supported by cross-sectional transmission electron microscopy (TEM) measurements (Fig. 4). Bright-field TEM images reveal that AlScN thin films grown in synchronization mode I have inter-columnar voids visible as bright areas extended along the growth direction (Fig. 4A), whereas the films grown using SFP-HiPIMS (mode III) have dense microstructure with no open grain boundaries (Fig. 4B). This is in agreement with the reduction of tensile residual stresses. The observed densification is a direct consequence of an ion-induced densification process, which becomes more efficient when the plasma-induced floating potential is synchronized with the Sc$^+$-rich portion of the pulse. Moreover, selected area electron diffraction (SAED) patterns reveal that the film grown in mode I is polycrystalline (Fig. 4C) whereas the one grown in SFP-HiPIMS mode III shows (002) out-of-plane texture and a diffraction pattern typical for single crystals (Fig. 4D). A high-resolution TEM (HR-TEM) image acquired from the film/substrate interface of the film grown in SFP-HiPIMS mode III (Fig. 4E) demonstrates hetero-epitaxy of the AlScN film on sapphire (001) with an out-of-plane AlScN [001]//Al$_2$O$_3$[001] and in-plane AlScN [100]//Al$_2$O$_3$[110] epitaxial relationship.

Overall, these results highlight the utility of SFP-HiPIMS for the deposition of highly-textured thin film materials at low deposition temperatures on insulating substrates. This is particularly interesting for the deposition of functional thin films on temperature sensitive substrates or device stacks. But the selective metal-ion acceleration enabled by SFP-HiPIMS is also useful for other applications. Analogous to the proof-of-principle experiment outlined in Fig. 2, HiPIMS depositions from two Ta targets were performed in Ar/N$_2$ atmosphere. By synchronizing one Ta pulse to the ToF of Ta$^+$ ions from the respective preceding Ta pulse, the hardness of the cubic TaN films was improved by almost 20%. (See Supplementary Information). This demonstrates the universal applicability of the concept.

## Discussion

This work introduces an original approach for preferentially accelerating specific ions during thin film deposition on insulating or electrically floating substrates−synchronized floating potential HiPIMS, or SFP-HiPIMS. The approach works on the principle of utilizing the negative floating potential generated during the HiPIMS pulse when the substrate is immersed in the sputter plasma. With appropriate synchronization, this potential can be used to accelerate specific ions onto the substrate. For the proof of concept, we investigated the effect of SFP-HiPIMS on the properties of AlScN films grown on a variety of insulating or electrically floating substrates. The setup involved sputtering from three magnetrons with two Al and one Sc target in a reactive environment and a confocal geometry. The Sc$^+$ ions were chosen to be accelerated due to their higher ToF and heavier mass, thus imparting more momentum through ion bombardment. The novel synchronization scheme leads to a significant improvement in the film quality. We were able to show that the ion bombardment with low kinetic energies around the lattice displacement threshold significantly improves the crystallinity, texture and stress state. For sapphire (001) substrates, it even facilitates epitaxial growth at low deposition temperatures, highlighting the potential of SFP-HiPIMS for the growth of high-quality thin films on insulating substrates at low substrate temperatures.

In conclusion, SFP-HiPIMS provides a solution for the tailored acceleration of ions onto insulating substrates, an important challenge in IPVD processes. In contrast to pulsed RF substrate biasing, where a certain degree of process-gas-ion acceleration is unavoidable, SFP-HiPIMS offers much more gentle deposition conditions. Therefore, this deposition scheme holds the promise of extending the use cases of IPVD techniques, such as HiPIMS, to defect-sensitive materials with the goal of improving film quality, lowering the overall processing temperatures, but also enabling unique or improved functionalities.

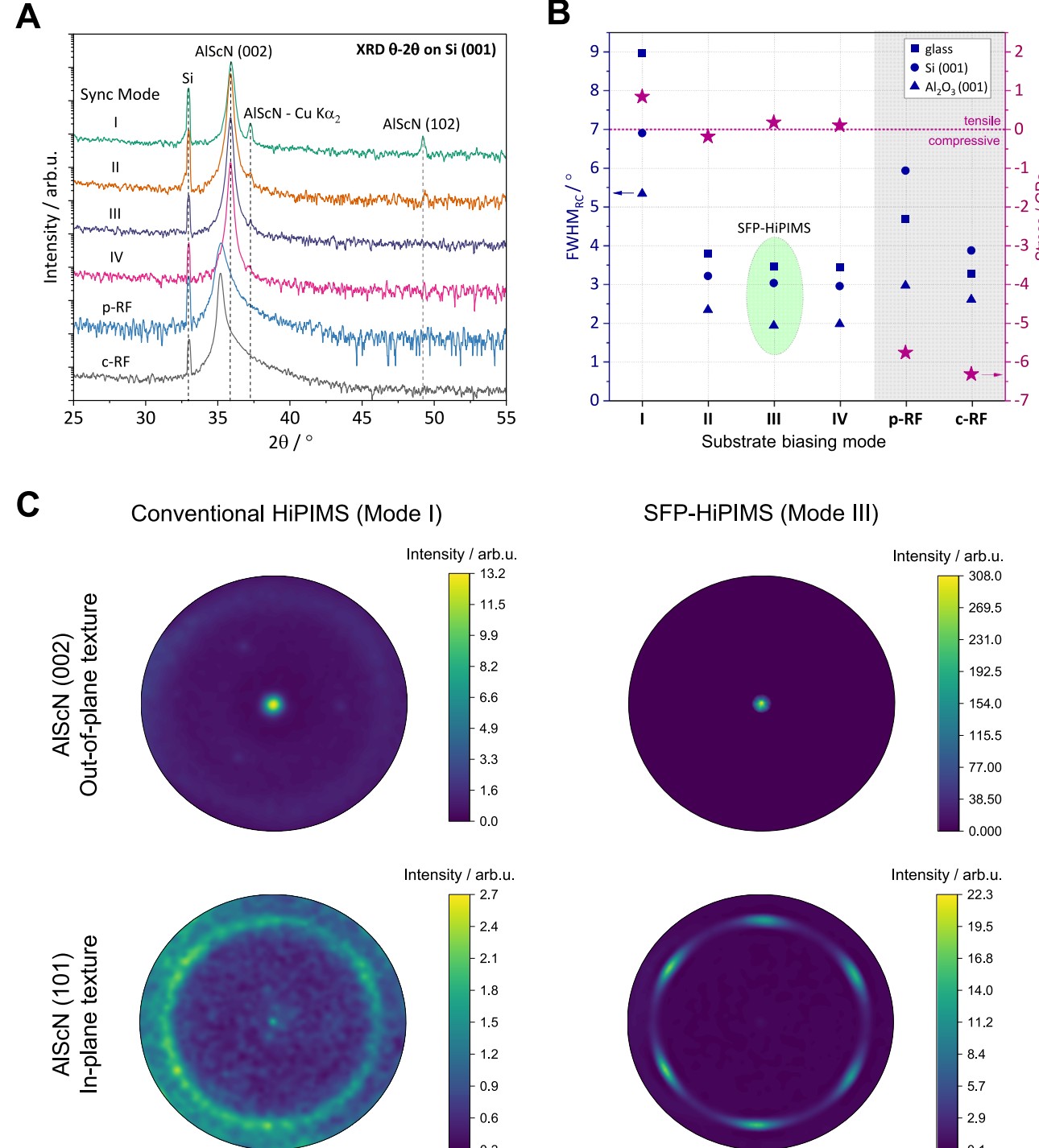

**Fig. 3 | Structural analysis and residual stress measurement for AlScN deposited on electrically floating substrates using different synchronization modes as well as pulsed and constant RF biasing (p-RF, c-RF).** A XRD measurements of AlScN films on Si (001). **B** FWHM of the AlScN (002) rocking curve (FWHM$_{RC}$) for different substrates and residual stress on Si (001). **C** Pole figure analyses of AlScN deposited on sapphire (001). It is apparent, that the synchronized floating potential significantly improves the crystallinity and texture but also reduces the tensile residual stress in the films for all tested substrates.

This is particularly relevant considering the increasing demands for sustainable manufacturing processes and increasingly temperature-sensitive device structures in emerging electronic applications. But also, other technologies benefit from the approach. For instance, in photonics or quantum acoustics where piezoelectric thin films are often deposited on thick insulating materials[35,36]. Here, SFP-HiPIMS can provide an advantage when compared with conventional deposition methods.

SFP-HiPIMS can be implemented for many materials and substrates and therefore can benefit various applications and technologies. Importantly, the approach can be implemented with little additional cost on most commercial sputter systems.

## Methods

The AlScN films were deposited using a custom-built AJA ATC-1800 chamber, with a base pressure of <10$^{-6}$ Pa. The films were deposited

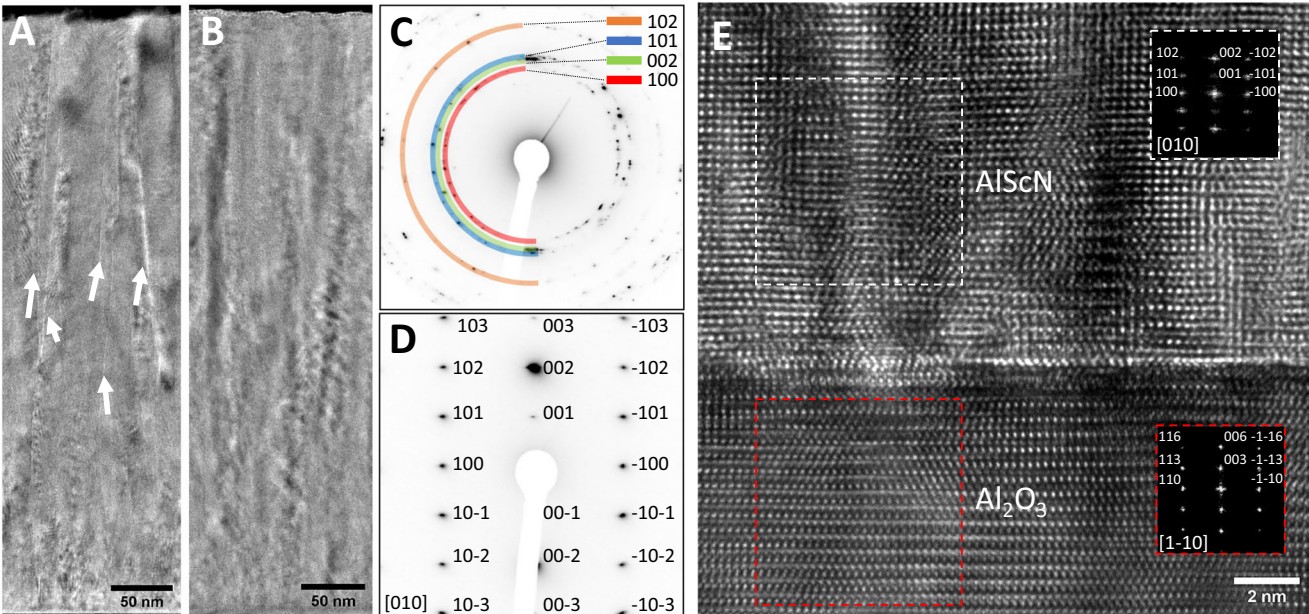

**Fig. 4 | Microstructure of AlScN thin films deposited on sapphire (001) substrates using conventional HiPIMS (Mode I) and SFP-HiPIMS (Mode III).** Bright-field transmission electron microscopy (BF-TEM) cross-sectional images of AlScN films grown using conventional HiPIMS show inter-columnar voids (**A**), while SFP-HiPIMS results in a compact microstructure (**B**). Selected area electron diffraction (SAED) patterns show random in-plane orientation for conventional HiPIMS (**C**) and epitaxial growth for SFP-HiPIMS (**D**). **E** HR-TEM image of the film/substrate interface for the SFP-HiPIMS deposition with fast Fourier transform (FFT) patterns as inserts.

on multiple substrates including p-type Si (001), Eagle-XG borosilicate glass and sapphire (001) (CrysTec AG). The type-2 unbalanced magnetrons in the chamber are equipped with 2 inch Al (Lesker, purity: 99.999 at.%) and Sc targets (Hunan Advanced Metal Materials, 99.999 at.%, O < 800 ppm) in confocal geometry. The substrates were put at a working distance of 12 cm and were heated from the back using 5 halogen lamps, ensuring uniform heat distribution across the whole sample holder. All depositions were performed at a pressure of 5 μbar, with the flow rates of Ar and $N_2$ in the chamber maintained at 20/12 (sccm/sccm). The depositions were performed at a substrate temperature 100 °C to maintain a constant, low temperature throughout the deposition. This temperature was calibrated prior to the deposition using a thermocouple mounted to the surface of a Si wafer. The substrates were cleaned in an ultrasonicator in acetone and ethanol individually, and further baked out at 300 °C in the deposition chamber prior to the deposition. The power to the sputtering targets was provided using Hipster 1 power supplies by Ionautics and the pulses were synchronized using an Ionautics synchronization unit. For the synchronization of the substrate's floating potential, the ToF of the Al and Sc ions was estimated by time-resolved mass spectrometry measurement using a Hiden Analytical EQP. The time-resolved measurements were performed for the relevant ionic species such as $^{36}Ar^+$ and $^{45}Sc^+$. A less abundant isotope of Ar is used here to avoid the saturation of the detector. The time-of-flight in the mass spectrometer is calibrated by applying a gating potential at the driven front end of the spectrometer. A more detailed description of this method can be found in our previous publication[11]. The substrate floating potential was measured using a passive probe with 10 MΩ impedance (Tektronix TPP0502) connected to the electrically isolated substrate holder (area approx. 100 cm²). To validate these measurements, a custom-built floating probe (area approx. 9 cm²) was positioned in the substrate position with no contact to the substrate holder or any of the chamber parts. Both floating potential measurements agreed almost perfectly (see Supplementary Fig. 2). During the depositions with Sync. Mode I-IV the substrate

holder was completely floating with no diagnostic equipment connected. For the depositions using the RF substrate bias a commercial 300 W RF power supply operating at a frequency of 13.56 MHz (Seren R301) was connected to the substrate. For the pulsed RF substrate biasing the same power supply was operated in pulse mode and the TTL trigger was supplied by the Ionautics synchronization unit. The same passive probe (Tektronix TPP0502) was used to measure the RF bias potential applied to the substrate as well as the resulting DC self-bias. Detailed information about the applied continuous and pulsed RF biases can be found in the supporting information (see Supplementary Fig. 5).

X-ray diffraction (XRD) analysis of the films was performed using a Bruker D8 in Bragg-Brentano geometry and Cu-kα radiation. The pole figures were obtained for the films for psi ranging from 0° to 80° with a step size of 3° in psi and phi. Each data point reported for the structural analysis represents measurements on a single physical sample. In addition to the data reported in the manuscript, the findings were reproduced several times with slightly varying deposition conditions and the general trends reported in the manuscript were consistent. One of these additional studies is also reported in the supporting information (Supplementary Fig. 10). For stress analysis, the measurements were performed on multiple positions of the wafer. The residual stress in the films was measured by estimating the curvature of the wafer before and after the deposition. For this, the Si (002) peak position was determined via rocking curve measurements in XRD at several positions on the wafer. The substrate curvature (R) was determined by calculating the slope of the plot of measurement positions (x) against the peak position of the rocking curve (θ). The curvature was then translated to stress (σ) with the help of Stoney's equation,

$$\sigma = \frac{1}{6} \frac{E_s}{1 - \nu_s} \frac{t_s^2}{t_f} \left( \frac{1}{R} - \frac{1}{R_o} \right) \quad (1)$$

Where $E_s$ and $\nu_s$ are the elastic modulus and Poisson's ratio for the substrate. For Si(001), the value of ($E_s$ / 1 - $\nu_s$) is 181 GPa at room

temperature[37]. $t_s$ (≈ 0.540 mm) and $t_f$ are the thickness of the substrate and film, and $R$ and $R_o$ are the curvature of the wafer before and after the deposition.

The thickness of the films was measured using a Bruker Dektak XT profilometer. The cation concentration was determined using X-ray fluorescence in a commercial system featuring Rh irradiation (Helmut Fischer SDV-SDD). The optics were flushed with He during the measurement to increase the sensitivity for light elements (i.e. Al). AFM images were acquired using Bruker Nanoscope in Scanasyst-Air mode using Si cantilevers with spring constant of 0.4 N/m paired with a 2-nm sharp tip. The images are further processed in Gwyddion software. Nanoindentation measurements are performed using a Picodentor HM500 equipped with a Berkovich diamond tip. The indentation depth is limited to less than 15 % of the total film thickness to minimize the elastic behavior from the substrate during nanoindentation. 30 indents are performed for each sample to determine the average value and standard deviation. Hardness was determined following Oliver and Pharr method[38,39].

A JEOL LEM-22000FS microscope operated at 200 kV was used for transmission electron microscopy analysis. TEM specimens were prepared by focused ion beam (FIB) lift-out procedure in a FEI Helios NanoLab G3 UC Dual Beam SEM/Ga+ FIB system.

The Ar-content in the films was measured using X-ray photoelectron spectroscopy (XPS) as well as Rutherford backscattering spectrometry (RBS) and elastic recoil detection analysis (ERDA). Details about these measurements are provided in the Supporting Information.

## Data availability

The data that support the findings of this study are openly available in an online repository (https://doi.org/10.6084/m9.figshare.28062617).

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

## Acknowledgements

J.P. and O.P. acknowledge funding from the SNF (projects 196980 and 227945). Arnold Müller from the Laboratory of Ion Beam Physics at ETH Zurich is gratefully acknowledged for performing RBS/ERDA analyses.

## Author contributions

J.P. and S.S. conceived the idea for the study and S.S. supervised the project. J.P. performed the thin film growth and characterization with support of O.P., K.T. and S.S. O.P. performed TEM analysis of selected samples. J.P. wrote the original draft of the manuscript with contributions from L.S. and S.S. All authors participated in review and editing of the article and approved its final version.

## Competing interests

The authors declare that a related patent application has been filed: Title: "Method and apparatus for depositing a material layer", application number: "EP 24163606", status: "pending". This patent covers the general concept of SFP-HiPIMS, especially the use of a transient substrate floating potential to accelerate film-forming ions. The applicant is Empa, the inventors are J.P. and S.S. The remaining authors declare no competing interests.
