## [Transparent Peer Review File · Nature Communications]

Low Temperature Deposition of Functional Thin Films on Insulating Substrates Enabled by Selective Ion Acceleration using Synchronized Floating Potential HiPIMS

Corresponding Author: Dr Sebastian Siol

Version 0:

Reviewer comments:

Reviewer #1

(Remarks to the Author)

The manuscript claims to introduce a novel method for ion acceleration towards insulating substrates using the floating potential. It involves the use of more than one sputtering cathode to synchronize ions to be accelerated with the floating potential established during the previous pulse of the first cathode.

In my opinion, the manuscript includes the following critical inconsistencies.

In the abstract is stated: "One key challenge however is, how such potential can be applied on insulating or electrically floating substrates."

Later, this challenge is answered in Introduction: "The state-of-the-art method for accelerating ions on an insulating substrate involves applying a radio frequency (RF) plasma to the substrate, thereby inducing a negative floating potential."

For diminishing or dismissing this possibility it is claimed that "RF biasing of insulating substrates, though effective, is limited to higher voltages, which can lead to a substantial amount of ion implantation during film growth, resulting in strain and undesirable defects.[20]"

However, this statement is not correct. The RF bias is not limited to high voltages and can easily be tuned even below 100 V. Ion implantation occurs for much large voltages. Yet even 100 eV is enough to cause preferential resputtering.

In Conclusion is claimed "Overall, SFP-HiPIMS provides a practical and low-cost solution for the tailored acceleration of ions onto insulating substrates, a fundamental challenge in ionized physical vapor deposition (IPVD) which had not been solved to date."

However, in Experimental section is stated: "For the synchronization of the substrate's floating potential, the ToF of the Al and Sc ions was first estimated by time-resolved mass spectrometry measurement using a Hiden Analytical EQP-300."

It means that for using this method one needs to estimate by a mass spectrometer the ToF for each of the deposition conditions. An approach that is neither low-cost nor practical for most sputtering systems.

Besides these aspects, the claimed synchronization is only briefly supported by Figure 2.

Reviewer #2

(Remarks to the Author)

The authors aim to control the microstructure of thin films, in particular nitride films such as AlScN, using an ion-selective deposition method involving two or more (here three) unbalanced magnetrons and insulating substrates. There are two challenges that need to be addressed: (i) knowing the time-depending plasma composition during each plasma pulse and (ii) how to control the floating potential that establishes itself on the surface of the insulating substrate such as glass or sapphire in order to tune ion acceleration.

The approach they chose is interesting: affecting the floating potential of the substrate surface by a plasma of one magnetron plasma and making use of such potential by a second plasma from a second (or third) unbalanced magnetron. The authors use the delay time between plasma pulses to demonstrate the desired effect.

While the general idea is interesting, and positive, desirable effects have been found, I think the work lacks important details that need to be shown and discussed. Most importantly, the effect is claimed to be associated with the time-depending

nature of the floating potential of the substrate surface. Measurements of the floating potential is shown in Fig. 2 but it is not clear how this was measured. The floating potential is the result of the balance of electron and ion currents to the floating (or insulating) surface. Given the high mobility of electrons, the characteristic time of change is associated with the time of change of such currents, and in the absence of plasma on the leakage current to a neighboring potential, e.g. of a grounded or biased substrate holder. This is not trivial since the effect lasting from the previous plasma pulse is likely to be quickly overwritten when the following plasma pulse arrives.

In the article file, I have made specific annotations. The most important I list here, too:

1. Abstract: when claiming synchronization of ion arrival and floating potential, the time-dependence of the floating potential must be mentioned or explained here, not just in the text. The claims currently made appear a bit strong.
2. In the Introduction, page 3 of the PDF, the claim of "avoiding" gas incorporation is too strong, I think. My assessment is also supported by a statement on page 11, namely that "a less abundant isotope of Ar is used here to avoid saturation of the detector." suggesting that there are plenty of Ar ions. Even as the plasma composition changes, it is to my knowledge never free of argon gas unless one operates in the special cases of very low pressure sputtering (W. M. Posadowski, *Vacuum* 46, 1017 (1995)), or gas-free sputtering (J. Andersson et al., *Appl. Phys. Lett.* 92, 221503 (2008)).
3. page 6 and related places: also here, the claim "In addition, the synchronization can be adjusted to specifically avoid the acceleration of undesirable ionic species, such as Ar⁺ or other process gas ions." is not really proven in the current article. Especially at this location, the text resembles of a text written for a patent application, where the process description needs to be plausible. Here, a higher standard must be applied. It is not directly proven that the acceleration of argon ions is avoided. There is evidence that the timing of the pulses has the desired effect on the film properties, yet, the underlying reasoning is not at the level of proof.
4. page 7, figure 2, contains the time-dependence of the floating potential; a revised version of the manuscript must clearly describe the how these data were acquired.
5. page 8, figure 3, the curve in subfigure B is misleading since on the left side it does not represent the average of the data points; the figure caption does not mention the meaning of the dashed line; additionally, this subfigure is not clear how many samples have been used to obtain the data. A minimum of 4, of course, but more than that? I wonder since the stress changes sign. Was that reproducible? Also, for most material systems, HiPIMS delivers films of compressive stress, not tensile, and so the data are somewhat surprising.
6. page 11: the authors report about substrate temperature of 100 C. How was that measured? Is there any information if the temperature went up during deposition?
7. page 11: Here is where a report in the floating potential measurements should be given.

Reviewer #3

(Remarks to the Author)

Thank you for letting me review this contribution on AlScN thin film deposition using HiPIMS. The authors describe a novel method for moderate ion acceleration by suitably timing the occurrence of a floating potential on the substrate. These findings are indeed important for the plasma-based thin film community, and I very much enjoyed reading the manuscript.

In general, I advise the authors to revise the introduction on floating substrate potential since it is key for the present work and there is some confusion concerning the floating potential and an RF self-bias potential in the introduction (see for example item 3 below). There is also a significant overlap between the introduction and Section 2.1 on this matter. Instead of using the RF discharge as a starting point, my suggestion is to use a more classical description of floating potential from classical textbooks in plasma physics and probe theory where the starting point is an electrically floating object in contact with a plasma (independent on how the plasma is generated) and the floating potential is defined as the potential assumed by a probe/surface when the net current collected by it is zero.

Another question that arises is how to tune the ion energy using this strategy? In your previous work (Ref 33 in the manuscript) you identify -30 V substrate bias as the best choice. In the present work the floating potential peaks at -20 V (or slightly below), which is also what is expected in these type of plasma discharges. How can one increase this ion-accelerating potential using the approach presented here?

I recommend the authors to revise the manuscript by taking these suggestions into account along with the detailed comments below.

Detailed comments:

1. Page 2: Use ionized flux fraction instead of ionization rate throughout the manuscript. A better (and more recent) overview of ionized flux fractions is given by Butler et al., *Plasma Sources Sci. Technol.* 27 (2018) 105005.
2. Page 3: Please revise the explanation of arrival times of different ionic species. Working gas ions do not arrive earlier to the substrate due to gas rarefaction. It is due to ionization of working gas in the plasma volume during the onset of the HiPIMS pulse. These gas ions will have a shorter travel distance to the substrate as compared to sputtered species from the target (which initially require gas ions to be sputtered in the first place). A detailed explanation is given by Gudmundsson et al., *J. Vac. Sci. Technol. A* 30(3) (2012) 030801.

3. Page 3: Revise description on RF biasing, where it is stated that such biasing induces a negative floating potential: Applying an RF bias onto a substrate will lead to the substrate attaining a (DC) self bias. Generally, we make a difference between a (DC) self bias and a floating bias. Towards the end of the introduction (page 4), the concept of floating bias is again brought up when introducing other sputtering techniques. I recommend that the authors introduce floating potential in a different way by not focusing on the sputtering techniques, but rather in the general sense (electrically insulating object in contact with a plasma). This will avoid any confusion of when a floating potential occurs.
4. Page 3: Why is pulsed (synchronized) RF bias limited to longer pulses? What timescales are we talking about? Hubicka and co-workers have shown pulsed RF bias synchronization on the same timescale as HiPIMS pulses. See for example: Hubicka et al., "Hardware and power management for high power impulse magnetron sputtering" High Power Impulse Magnetron Sputtering High Power Impulse Magnetron Sputtering Fundamentals Technologies Challenges and Application, edited by D. Lundin, T. Minea, and J. T. Gudmundsson (Elsevier, Amsterdam, 2020).
5. Figure 1 A: Use a different color for the potential profile as to not mix it up with the cathode/anode (black color).
6. Page 5: From a plasma physicists' perspective, it is not clear how the authors connect the plasma potential to plasma stability. This may lead the reader to think in terms of plasma instabilities, which is not at all what this is about. The plasma potential comes from the theory on plasma sheaths and the need for the plasma to reduce the loss of electrons to maintain quasineutrality.
7. Page 5: The plasma is not generated instantaneously. There is a period of plasma expansion during which the floating potential changes significantly. The description here is misleading and also gives the impression that the floating potential is constant (see for example Fig. 2).
8. Page 7: The delta in Δ_t is commonly associated with a time interval. The authors use Δ_t when referring to a time instant. This might confuse the reader into thinking that different pulse lengths were used.
9. Figure 2: I cannot understand why the authors see little to no argon during the Al-HiPIMS pulses (and no difference when delaying the Al-HiPIMS pulse). How was this mass spec measurement carried out?
10. Pages 7-8: It would be of great interest to know the Ar content in the deposited samples, i.e., how it varies vs delay time.

Version 1:

Reviewer comments:

Reviewer #1

(Remarks to the Author)

The authors made a considerable improvement of their manuscript by including additional measurements with RF bias, remeasuring the floating potential and by addressing all received comments. In my opinion, the results are method oriented and I still disagree with the following statement included in the abstract: "The results of this study demonstrate that SFP-HiPIMS provides a practical and economical solution for a long-standing challenge in physical vapor deposition, which can be implemented in standard deposition equipment." TOF measurements are not easy and one can expect significant deviations when changing the experimental arrangement. HiPIMS operated at low pressures generates already very energetic species able to compact the films. In conclusion I see a certain impact but less moderate than claimed.

Reviewer #2

(Remarks to the Author)

Report on revised version of NCOMMS-24-52890 by Siol et al.

The authors have done a good job in responding to the reviewers' critical points, and the additional material is valuable. While I think the work deserves publication because it contains interesting results, I'm still not convinced that the synchronization claim in its current form is valid. I base this primarily on a point made by me and perhaps even clearer by reviewer 3, namely that the establishment of the floating potential is the result of the sheath physics: the floating potential is the result of the balance of electrons and ions arriving at the surface leading to net zero current, or, as reviewer 3 put it, to reduce the loss of electrons to maintain quasineutrality. Either way, the floating potential does not "linger" for much time as it continuously adjust itself to the plasma and sheath conditions. The wording "The floating potential is mostly constrained to the duration of the HiPIMS pulse and occurs almost immediately when the sputter pulse is applied." is on the one hand trivial because when the plasma is present, the surface potential shifts automatically to a negative value due electrons having a higher mobility than ions and the requirement to maintain net zero current for an insulating surface. The wording is also not quite appropriate because the surface has always a potential. The floating potential (of a surface) does not appear and then disappears. What the authors mean is that the surface potential shifts to negative values in the presence of plasma relative to this plasma. The observation that a negative floating potential appears during the plasma pulse is well known. The correlation of plasma presence and negative floating potential is not sufficiently well shown in Fig. 1 because the sheath in front of the substrate is missing and therefore there is no correct display of the potential distribution. The effect seen, however, is real, and I believe that a possible explanation, qualitatively in line with the authors, is that the

arrival of the metal ions of magnetron 1 is timed when there are rather hot electrons from magnetron 2. The latter gives rise to a negative surface potential and that accelerates whatever ions are present at this time at the substrate sheath edge. The acceleration per is not selective of specific ions but ALL ions present at the sheath edge are accelerated. Since the plasma composition is time-dependent, one may achieve selectivity by timing the plasma composition. This has been shown by timing applied bias pulse in relation the HiPIMS pulse [G. Greczynski, et al., J. Appl. Phys. 121, 171907 (2017)]. Here, an attempt is made to achieve a similar effect for an insulating substrate by using more than one pulsed magnetron. In other words, the authors have made interesting experiments and showed interesting and beneficial effects, the explanation however falls short because the sheath physics above the substrate surface is not properly explained.

Another remaining point of criticism is related to RF bias versus RF discharge. (“...achieving stable ignition of an RF plasma for substrate biasing”). When plasma is provided, e.g. by the HiPIMS discharge, there is no need to also ignite an RF discharge to get an effect of RF bias. RF bias is supposed to affect the surface potential of the substrate, not to generate ions in an RF discharge. I think the concept of (pulsed) RF bias and (pulsed) RF discharges are mixed up in the description. Both points of criticism are related because they indicate that the physics of the substrate sheath is not considered as needed.

Reviewer #3

(Remarks to the Author)

I highly appreciate the efforts the authors have taken in revising the manuscript. Their work, particularly in carrying out additional experiments on floating potential and RF biasing, adds significant value to the study (see also comments below). The manuscript is now much more complete, and I commend them for their thorough revisions.

Additionally, the authors have addressed all the questions I raised in the previous round, and I am satisfied with their responses. Based on the improvements made, I recommend the manuscript for publication. The authors may consider the following minor comments:

1. Floating potential: The authors have taken great care to revise their introduction on floating potentials. I highly appreciate their added efforts and new measurements of the floating potential in their HiPIMS discharges.

Page 4, top: Change to ‘electrically isolated substrate’ instead of only substrate.

Regarding my previous question on tuning the ion energy using their strategy, the authors suggest that a wider range of the floating potential could be achieved by adjusting the balancing of the magnetron or modifying the deposition geometry. While these methods may indeed work, I am not aware of any significant changes in the floating potential resulting from such adjustments (as also shown in the supplementary material). However, the authors demonstrate that very high bias values are not necessary (not even desired) for the current material system, so this concern is less relevant in this context and can be considered outside the scope of the present study.

2. RF biasing: The authors have done an excellent job revising the introduction on RF substrate biasing. Additionally, they have successfully carried out further experiments to compare their approach (SFP-HiPIMS) with conventional HiPIMS + RF substrate biasing, which adds valuable insight to the work. It is very exciting to see that their SFP-HiPIMS process shows a clear advantage.

Page 4, middle: Another issue of using RF biasing in HiPIMS is the difficulty of maintaining a constant self-bias, which ultimately will affect the ion acceleration. In my opinion, this is even more problematic than generating a low-density substrate argon plasma.

Page 9: How were the RF self-bias values calculated in Figure S4? From Figure S5 it is seen that the self-bias amplitude changes in time. Did the authors pick a time-averaged value?

Page 11, bottom-half: Thank you for also measuring the Ar⁺ ion energies when using RF biasing. This is a really valuable contribution. I would, however, add to the explanation on page 11 that the higher Ar⁺ ion energies observed during RF biasing is likely due to the high accelerating voltages seen on the oscilloscope viewgraphs of the RF bias (Figure S5), where the self-bias voltage is not constant but rather increasing throughout the entire pulsed RF bias period. It is therefore very likely that a fraction of ions will be accelerated to higher energies than desired.

Version 2:

Reviewer comments:

Reviewer #1

(Remarks to the Author)

Thank you for addressing the comments.

Reviewer #2

(Remarks to the Author)

The second round of revision did bring the required / expected clarity of explanation to the observed results. This manuscript - in its current form - is a very nice piece of work that is ready for publication. All comments have been taken into account and I can recommend publication.

Preface – Answer to the reviewer comments for NCOMMS-24-52890

We thank all the reviewers for their thorough constructive criticism and feedback. This feedback has been instrumental in improving our manuscript. We were able to address all points in the revised version of our manuscript. The major changes are highlighted below and referenced in the respective answers to the individual reviewers:

1. Corrected the measured floating potential and demonstrate tunability of the same

When trying to reproduce and confirm our previous measurements of the floating potential we noticed a discrepancy between the data. We realized that the old measurements had unintentionally been performed with a $50\ \Omega$ feedthrough terminator attached to the substrate. This effectively reduced some of the measured floating potential. The values for the floating potential reported in the original draft of the manuscript were therefore too low. The depositions were unaffected by this, since no diagnostic equipment was attached during the depositions and the substrate was always fully floating.

The measurements were repeated with a high impedance passive probe ($10\ \text{M}\Omega$) which revealed higher overall floating potentials. These values were further confirmed by additional measurements using a smaller custom-build floating potential probe which was placed in the substrate position.

We have updated the method section to give more detail on the measurement of the floating potential. In addition, we added additional information in the supporting information comparing floating potential measurements on both the substrate holder and the custom floating potential probe (**Figure S2**).

In addition, we show that the floating potential can be tuned by changing the discharge voltage (at constant power) of the accelerating HiPIMS pulse (**Figure S3**).

2. Inclusion of samples deposited with continuous RF bias and pulsed RF bias

A critique present in all reviews concerned the comparison of SFP-HiPIMS to the current state of the art: RF substrate biasing. When starting the project, based on our own experience in applying RF substrate biasing during DCMS, as well as the limited reports in literature we operated under the assumption, that RF substrate

biasing at very low voltages and pulse length was not feasible. Motivated by the reviewer comments we implemented suitable experimental capabilities in our laboratory to test and benchmark RF substrate biasing under these conditions ourselves.

Importantly we find, that in the presence of a HiPIMS discharge, the RF substrate plasma gets stabilized to powers as low as 1–2 W. In addition, a stable pulsed operation on the order of a few μs is possible. We now report the boundary conditions for pulsed RF substrate biasing in our deposition tool in the supporting information (**Figure S4**). We believe that this information will also be helpful for other researchers in the community. The corresponding statements regarding the limited stability window of the RF substrate plasma have been removed from the manuscript.

Using these newly implemented capabilities, we performed reference depositions using both a continuous RF substrate (c-RF) bias as well as a pulsed RF substrate bias (p-RF) generating a -30 V DC self-bias, synchronized to the metal-ion arrival at the substrate (**Figure S5**).

Strikingly, despite comparable substrate bias potentials, the quality of the films deposited with RF substrate biasing is not comparable to those deposited using SFP-HiPIMS (**Figure 3** below).

We find that RF substrate biasing leads to significant Ar incorporation on the order of 1 at.% revealed by XPS and RBS/ERDA (**Figure S6/Figure S7**). We demonstrate that the reason for this difference in film quality is rooted in the higher Ar-ion energy in the RF case, when compared to the SFP-HiPIMS deposition. RFEA analysis reveals that in order to achieve a DC self-bias of -31 V, Ar-ion energies of over 60 eV are measurable in the RF substrate plasma (**Figure S8**). These energies are well above the lattice displacement energy and consequently lead to defects in the film. Since the substrate RF plasma is formed mostly from the process gas the acceleration of the corresponding ions is hard to avoid.

This is a very important result, especially for the deposition of defect-sensitive materials with moderate lattice displacement energies like wurtzite AlN. Here, the *SFP-HiPIMS* process shows a clear advantage with its ability to accelerate metal-ions while simultaneously minimizing detrimental process gas ion acceleration.

REVIEWER COMMENTS

Reviewer #1 (Remarks to the Author):

The manuscript claims to introduce a novel method for ion acceleration towards insulating substrates using the floating potential. It involves the use of more than one sputtering cathode to synchronize ions to be accelerated with the floating potential established during the previous pulse of the first cathode. In my opinion, the manuscript includes the following critical inconsistencies.

In the abstract is stated: "One key challenge however is, how such potential can be applied on insulating or electrically floating substrates." Later, this challenge is answered in Introduction: "The state-of-the-art method for accelerating ions on an insulating substrate involves applying a radio frequency (RF) plasma to the substrate, thereby inducing a negative floating potential."

For diminishing or dismissing this possibility it is claimed that "RF biasing of insulating substrates, though effective, is limited to higher voltages, which can lead to a substantial amount of ion implantation during film growth, resulting in strain and undesirable defects.[20]"

However, this statement is not correct. The RF bias is not limited to high voltages and can easily be tuned even below 100 V. Ion implantation occurs for much large voltages. Yet even 100 eV is enough to cause preferential resputtering.

Answer:

We thank the reviewer for their valuable feedback. This motivated us to perform additional experiments that significantly improved the quality of the manuscript and deepened our understanding of the limitations of RF substrate biasing for the deposition of defect sensitive materials.

In previous studies we found, that kinetic energies around the cation lattice displacement energy (LDE) are ideal to produce low-defect films while still facilitating significant improvements in crystallinity and texture. At the same time it is important to avoid Ar-ion bombardment at energies much higher than the LDE. This means that acceleration potentials on the order of -20 V to -30 V are desirable. In previous studies with DC magnetron sputtering in our own group, we found that at typical sputter pressures the lowest achievable RF substrate bias was over 50 V. In addition, such low RF substrate biases are rarely reported in literature. This led us to believe that controlling RF substrate bias below these values is challenging.

However, following the feedback of the reviewer, we revisited this topic and explored the limits of RF substrate biasing for different synthesis conditions. We found that the presence of the HiPIMS plasma allowed us to ignite and sustain the RF plasma at significantly lower powers and pressures than previously anticipated. Specifically, we observed that a stable RF plasma could be maintained at pressures as low as 5 μ bar and 1 W of RF power, yielding a minimum obtained average bias of -28 V, close to our targeted acceleration potential. With this understanding, we successfully reconfigured our RF power supplies to deliver synchronized pulsed RF plasma.

These newly established experimental capabilities allowed us to perform depositions using RF substrate biasing to conduct a direct comparison of the two ion-acceleration strategies. We synthesized AlScN films with 1) a low constant RF substrate bias and 2) a pulsed RF substrate bias potential with timing and magnitude comparable to the SFP process reported in our work.

Strikingly, we find that the films deposited using the RF substrate bias potentials show excess compressive stress and measurable Ar incorporation (see Fig. 3, Fig. S6, Fig. S7, Table S1). These findings have been confirmed using RBS/ERDA analyses measurements as well as wafer curvature analysis on films deposited on electronically floating Si substrates. The inferior quality of the films produced with RF substrate bias can be explained by the Ar-ion bombardment during the application of the RF substrate bias. We measured the time-averaged ion energy distribution function of the RF discharge using a retarding field energy analyzer (RFEA) and found that for a DC self-bias around -31 V the Ar ions striking the substrate exhibit kinetic energies of up to 60 eV, well above the lattice displacement energy (Fig. S8). The thermalized Ar ions, which are always present during the HiPIMS process, on the other hand exhibit significantly lower energies on the order of only a few eV. Even during the HiPIMS pulse, when these thermalized ions are accelerated towards the substrate by the floating potential the resulting Ar-ion energy is much lower than during RF substrate biasing.

These are important results with significant impact for all researchers performing ionized PVD. It shows that there are significant limitations to the application of RF substrate bias potentials, especially when synthesizing defect sensitive materials. Our results demonstrate that for a given, desired acceleration of specific film-forming species, there is a simultaneous acceleration of Ar-ions from the RF plasma to similar or higher energies, effectively reducing the beneficial effect of metal-ion synchronized substrate biasing approaches.

Overall, these results highlight that SFP HiPIMS offers significant advantages compared to ion acceleration using RF substrate bias potentials.

Performed changes:

- We have thoroughly revised the abstract and introduction of the manuscript to highlight these limitations of RF substrate biasing for the growth of defect-sensitive materials.

- Added characterization data for the films synthesized with 1) constant RF substrate bias and 2) synchronized RF substrate bias.
- Added additional information about the lower boundaries of RF substrate biasing in different deposition conditions.
- Added additional details regarding the performed synthesis using RF substrate biasing (including RFEA analysis of the time averaged Ar ion energy distributions).

In Conclusion is claimed "Overall, SFP-HiPIMS provides a practical and low-cost solution for the tailored acceleration of ions onto insulating substrates, a fundamental challenge in ionized physical vapor deposition (IPVD) which had not been solved to date."

However, in Experimental section is stated: "For the synchronization of the substrate's floating potential, the ToF of the Al and Sc ions was first estimated by time-resolved mass spectrometry measurement using a Hiden Analytical EQP-300."

It means that for using this method one needs to estimate by a mass spectrometer the ToF for each of the deposition conditions. An approach that is neither low-cost nor practical for most sputtering systems. Besides these aspects, the claimed synchronization is only briefly supported by Figure 2.

Answer:

We thank the reviewer for this comment. We appreciate the raised concerns about the practicality and cost of using time-resolved mass spectrometry to determine the time of flight (ToF) of ions. While this was used in our study, we have clarified in the manuscript that the ToF can also be measured using more economical instruments like retarding field energy analyzers (RFEA). RFEA button probes are available at a fraction of the cost and commonly used in academic and industrial process development. Although RFEA lacks mass resolution, it can often reliably distinguish between metal ions and process gas ions, as demonstrated in prior studies. Additionally, we note that the ToF is not highly sensitive to smaller changes in process parameters. In addition, our results show that even a non-ideal synchronization offers significant benefits. The wide parameter window allows rough estimations based on literature values or trial and error (by measuring film's properties) to implement SFP-HiPIMS without specialized equipment. This flexibility ensures that the technique remains accessible and practical for a broader audience.

Regarding measurements supporting the successful synchronization we see a clear overlap of the floating potential and the arrival of the Sc ions. These measurements are highly robust and reproducible. In addition, the films show a reduction in tensile stress indicative of ion bombardment and densification. At the same time there is no measurable Ar content from both XPS and RBS/ERDA analyses, which further supports a successful metal-ion acceleration.

Reviewer #2 (Remarks to the Author):

The authors aim to control the microstructure of thin films, in particular nitride films such as AlScN, using an ion-selective deposition method involving two or more (here three) unbalanced magnetrons and insulating substrates. There are two challenges that need to be addressed: (i) knowing the time-depending plasma composition during each plasma pulse and (ii) how to control the floating potential that establishes itself on the surface of the insulating substrate such as glass or sapphire in order to tune ion acceleration.

The approach they chose is interesting: affecting the floating potential of the substrate surface by a plasma of one magnetron plasma and making use of such potential by a second plasma from a second (or third) unbalanced magnetron. The authors use the delay time between plasma pulses to demonstrate the desired effect.

While the general idea is interesting, and positive, desirable effects have been found, I think the work lacks important details that need to be shown and discussed. Most importantly, the effect is claimed to be associated with the time-depending nature of the floating potential of the substrate surface. Measurements of the floating potential is shown in Fig. 2 but it is not clear how this was measured. The floating potential is the result of the balance of electron and ion currents to the floating (or insulating) surface. Given the high mobility of electrons, the characteristic time of change is associated with the time of change of such currents, and in the absence of plasma on the leakage current to a neighboring potential, e.g. of a grounded or biased substrate holder. This is not trivial since the effect lasting from the previous plasma pulse is likely to be quickly overwritten when the following plasma pulse arrives.

In the article file, I have made specific annotations. The most important I list here, too:

1. Abstract: when claiming synchronization of ion arrival and floating potential, the time-dependence of the floating potential must be mentioned or explained here, not just in the text. The claims currently made appear a bit strong.

We thank the reviewer for their suggestions. The statement has now been modified as below, improving the clarity about time-dependence of floating potential:

"The floating potential is mostly constrained to the duration of the HiPIMS pulse and occurs almost immediately when the sputter pulse is applied. Here we demonstrate that this transient negative potential can be used to accelerate positively charged ions onto the substrate. By synchronizing the ion arrival with the substrate's floating potential, specific ions can be accelerated preferentially, thereby enhancing adatom mobility and improving film quality while reducing the detrimental effects of energetic Ar⁺ ion bombardment."

2. In the Introduction, page 3 of the PDF, the claim of “avoiding” gas incorporation is too strong, I think. My assessment is also supported by a statement on page 11, namely that “a less abundant isotope of Ar is used here to avoid saturation of the detector.” suggesting that there are plenty of Ar ions. Even as the plasma composition changes, it is to my knowledge never free of argon gas unless one operates in the special cases of very low pressure sputtering (W. M. Posadowski, *Vacuum* 46, 1017 (1995)), or gas-free sputtering (J. Andersson et al., *Appl. Phys. Lett.* 92, 221503 (2008)).

We thank the reviewer for their comment. We fully agree with this statement, as there are always some thermalized argon ions with comparably low kinetic energy of only a few eV. During sputtering from unbalanced magnetrons, it is not possible to avoid exposure of the substrate to these Ar ions. The Ar ion flux originating from the HiPIMS discharge has significantly higher kinetic energy than the thermalized Ar ions which are present between the HiPIMS pulses. With the synchronization of the negative substrate bias, we are trying to avoid any further acceleration of these energetic Ar ions. The claims made are modified in several places throughout the manuscript to clarify this.

The explanation for the different time of flight (ToF) of ions, as suggested by reviewer is also modified, as follows:

"During HiPIMS, for each pulse, the process gas ions tend to arrive earlier at the substrate than the corresponding metal ions. This is primarily due the ionization of working gas within the plasma volume during the onset of the HiPIMS pulse. These working gas ions have a shorter travel distance compared to the sputtered species, which must first be ejected from the target before reaching the substrate.^{10m}

3. page 6 and related places: also here, the claim “In addition, the synchronization can be adjusted to specifically avoid the acceleration of undesirable ionic species, such as Ar⁺ or other process gas ions.” is not really proven in the current article. Especially at this location, the text resembles of a text written for a patent application, where the process description needs to be plausible. Here, a higher standard must be applied. It is not directly proven that the acceleration of argon ions is avoided. There is evidence that the timing of the pulses has the desired effect on the film properties, yet, the underlying reasoning is not at the level of proof.

In the revised version of the manuscript, we provide a detailed analysis of the Ar-content in all films. The SFP-HiPIMS show no measurable Ar content via RBS/ERDA and XPS (**Figure S6** and **Figure S7**). In addition, the film properties are best, when the timing of the floating potential is optimally matched to the metal-ion arrival at the substrate. These measurements further support that preferential metal-ion acceleration is achievable during SFP-HiPIMS.

4. page 7, figure 2, contains the time-dependence of the floating potential; a revised version of the manuscript must clearly describe the how these data were acquired.

We thank the reviewer for their constructive feedback. We now include detailed information on the measurement of the floating potential in the methods section with additional information in the SI.

"The substrate floating potential was measured using a passive probe with 10 M Ω impedance (Tektronix TPP0502) connected to the electrically isolated substrate holder (area approx. 100 cm²). To validate these measurements, a custom-built floating probe (area approx. 9 cm²) was positioned in the substrate position with no contact to the substrate holder or any of the chamber parts. Both floating potential measurements agreed almost perfectly (see **Figure S2**)."

5. page 8, figure 3, the curve in subfigure B is misleading since on the left side it does not represent the average of the data points; the figure caption does not mention the meaning of the dashed line; additionally, this subfigure is not clear how many samples have been used to obtain the data. A minimum of 4, of course, but more than that? I wonder since the stress changes sign. Was that reproducible? Also, for most material systems, HiPIMS delivers films of compressive stress, not tensile, and so the data are somewhat surprising.

We thank the reviewer for their critical feedback. We synthesized 3 samples per deposition condition. The FWHM values on the left axis in subfigure B represent the FWHM of individual samples rather than the average of multiple samples. For reproducibility, additional depositions were carried out without any intentional substrate heating. These results are now included in the SI (**Figure S10**). The stress values were calculated by measuring the wafer curvature before and after deposition on several locations.

The dashed line in subfigure B was included as a guide to the eye. It has been removed to avoid any confusions.

We acknowledge the reviewer's observation that HiPIMS-deposited films in the literature predominantly exhibit compressive stress. The tensile stress observed for the conventional HiPIMS deposition seems unusual. However, it is a common observation for AlScN. Alloying of Sc into the cation lattice leads to a distortion of the wurtzite structure that has been shown to lead to significant tensile stress. This has been previously reported by other groups using DC and pDC sputtering. We have observed this ourselves in dozens of samples and the effects scale with the amount of Sc introduced.

We have added an additional statement to clarify this:

"Without any deliberate ion acceleration, the AlScN films exhibit tensile residual stress. This is a common observation in this material as Sc-doping leads to a distortion of the wurtzite lattice^{12,33} As expected, the

energetic Sc-ion bombardment during SFP-HiPIMS leads to a reduction in tensile residual stress from 850 MPa to an almost stress-free state."

6. page 11: the authors report about substrate temperature of 100 C. How was that measured? Is there any information if the temperature went up during deposition?

We thank the reviewer for their comment. The substrate temperature was calibrated using a thermocouple placed on a Si wafer positioned over the substrate holder. This allowed us to measure the actual surface temperature of the substrate for various heater setpoints. Based on this calibration, the appropriate heater setpoint was selected to maintain the desired temperature. The thermocouple in the heater measures the temperature of the substrate holder from the back during the deposition. As the substrate heats up during the deposition the active heating is reduced to keep the temperature of the holder more or less constant. This ability to maintain a more constant substrate temperature throughout the process is the main motivation in choosing 100°C rather than no active heating.

We added the following statement to the method section to clarify:

"The depositions were performed at a substrate temperature 100°C to maintain a constant, low temperature throughout the deposition. This temperature was calibrated prior to the deposition using a thermocouple mounted to the surface of a Si wafer."

7. page 11: Here is where a report in the floating potential measurements should be given.

We thank the reviewer for their comment. As mentioned in one of the previous answers, the method section now includes, how the floating potential was measured, as follows:

"The substrate floating potential was measured using a passive probe with 10 MΩ impedance (Tektronix TPP0502) connected to the electrically isolated substrate holder (area approx. 100 cm²). To validate these measurements, a custom-built floating probe (area approx. 9 cm²) was positioned in the substrate position with no contact to the substrate holder or any of the chamber parts. Both floating potential measurements agreed almost perfectly (see **Figure S2**)."

Reviewer #3 (Remarks to the Author):

Thank you for letting me review this contribution on AlScN thin film deposition using HiPIMS. The authors describe a novel method for moderate ion acceleration by suitably timing the occurrence of a floating potential on the substrate. These findings are indeed important for the plasma-based thin film community, and I very much enjoyed reading the manuscript.

In general, I advise the authors to revise the introduction on floating substrate potential since it is key for the present work and there is some confusion concerning the floating potential and an RF self-bias potential in the introduction (see for example item 3 below). There is also a significant overlap between the introduction and Section 2.1 on this matter. Instead of using the RF discharge as a starting point, my suggestion is to use a more classical description of floating potential from classical textbooks in plasma physics and probe theory where the starting point is an electrically floating object in contact with a plasma (independent on how the plasma is generated) and the floating potential is defined as the potential assumed by a probe/surface when the net current collected by it is zero.

Another question that arises is how to tune the ion energy using this strategy? In your previous work (Ref 33 in the manuscript) you identify -30 V substrate bias as the best choice. In the present work the floating potential peaks at -20 V (or slightly below), which is also what is expected in these type of plasma discharges. How can one increase this ion-accelerating potential using the approach presented here?

We thank the reviewer for their valuable feedback and have revised the introduction of the manuscript accordingly. Most importantly we found that we had made a mistake in our previous measurement of the floating potential. During the measurement a 50 Ω feedthrough terminator attached to the substrate. This effectively reduced some of the measured floating potential. The values for the floating potential reported in the original draft of the manuscript were therefore too low. The depositions were not affected by this. We repeated the measurements with a high impedance passive probe on both the substrate holder and a custom-built floating potential probe (see **Figure S2**). The refined plasma potential present during the deposition was rather on the order of 30-35 V. In addition, we show that this potential can be tuned by changing the pulse parameters of the accelerating pulse. By changing the discharge voltage, we were able to adjust the floating potential between approximately 20-30 V using a single Al magnetron operating at 100 W average power (**Figure S3**). A wider range should be accessible when adjusting the balancing of the magnetron or adjusting the deposition geometry.

I recommend the authors to revise the manuscript by taking these suggestions into account along with the detailed comments below.

Detailed comments:

1. Page 2: Use ionized flux fraction instead of ionization rate throughout the manuscript. A better (and more recent) overview of ionized flux fractions is given by Butler et al., Plasma Sources Sci. Technol. 27 (2018) 105005.

We thank the reviewer for their comment. We have replaced the term "ionization rate" with "ionized flux fraction" throughout the manuscript to provide a more accurate description of the ionization process. We also appreciate the additional references suggested by the reviewer, as they contribute to enhancing the quality and depth of this work.

2. Page 3: Please revise the explanation of arrival times of different ionic species. Working gas ions do not arrive earlier to the substrate due to gas rarefaction. It is due to ionization of working gas in the plasma volume during the onset of the HiPIMS pulse. These gas ions will have a shorter travel distance to the substrate as compared to sputtered species from the target (which initially require gas ions to be sputtered in the first place). A detailed explanation is given by Gudmundsson et al., J. Vac. Sci. Technol. A 30(3) (2012) 030801.

We thank the author for their valuable suggestion. The explanation for different time-of-flight of ions has been revised now as follows:

"During HiPIMS, for each pulse, the process gas ions tend to arrive earlier at the substrate than the corresponding metal ions. This is primarily due the ionization of working gas within the plasma volume during the onset of the HiPIMS pulse. These working gas ions have a shorter travel distance compared to the sputtered species, which must first be ejected from the target before reaching the substrate.^{10"}

3. Page 3: Revise description on RF biasing, where it is stated that such biasing induces a negative floating potential: Applying an RF bias onto a substrate will lead to the substrate attaining a (DC) self bias. Generally, we make a difference between a (DC) self bias and a floating bias. Towards the end of the introduction (page 4), the concept of floating bias is again brought up when introducing other sputtering techniques. I recommend that the authors introduce floating potential in a different way by not focusing on the sputtering techniques, but rather in the general sense (electrically insulating object in contact with a plasma). This will avoid any confusion of when a floating potential occurs.

We thank the author for their valuable suggestion. We have revised the introduction accordingly to better introduce when a floating potential occurs and also to distinguish the two concepts "floating potential" and "DC self bias".

4. Page 3: Why is pulsed (synchronized) RF bias limited to longer pulses? What timescales are we talking about? Hubicka and co-workers have shown pulsed RF bias synchronization on the same timescale as HiPIMS pulses. See for example: Hubicka et al., "Hardware and power management for high power impulse magnetron sputtering" High Power Impulse Magnetron Sputtering High Power Impulse Magnetron Sputtering Fundamentals Technologies Challenges and Application, edited by D. Lundin, T. Minea, and J. T. Gudmundsson (Elsevier, Amsterdam, 2020).

We thank the reviewer for their valuable feedback. When starting the project, based on our own experience in applying RF substrate biasing during DCMS, as well as the limited reports in literature we operated under the assumption, that RF substrate biasing at very low voltages and pulse length was not feasible. Motivated by the reviewer comments we implemented suitable experimental capabilities in our laboratory to test and benchmark RF substrate biasing under these conditions ourselves. We find that RF substrate biasing in our chamber is facilitated by the presence of a HiPIMS plasma. Under these conditions RF powers of 1-2 W were feasible. The shortest pulses we could reliably generate were on the order of only a few μs . It is noteworthy that only one make and model of our RF power supplies was able to sustain a stable DC self-bias during HiPIMS sputtering (Seren R301). We plan to further investigate this in the future.

Since we were able to deposit films using continuous as well as pulsed RF substrate bias potentials with this new experimental setup, we synthesized reference samples and benchmarked the SFP-HiPIMS approach against films deposited using RF substrate biasing.

In summary we find, that the SFP-HiPIMS process leads to much lower overall Ar-ion energies and consequently better film quality. This is due to the unavoidable Ar-ion acceleration during RF substrate biasing.

In our case we found, that, to achieve a DC self-bias of -31 V, Ar ions with energy up to 60 eV were generated. This led to measurable Ar ion incorporation in the film and corresponding high compressive stress values.

5. Figure 1 A: Use a different color for the potential profile as to not mix it up with the cathode/anode (black color).

We thank the reviewer for their nice suggestion. The potential profile color has been changed and revised figure is added in the manuscript, as shown below:

[Figure Redacted]

6. Page 5: From a plasma physicists' perspective, it is not clear how the authors connect the plasma potential to plasma stability. This may lead the reader to think in terms of plasma instabilities, which is not at all what this is about. The plasma potential comes from the theory on plasma sheaths and the need for the plasma to reduce the loss of electrons to maintain quasineutrality.

We thank the reviewer for highlighting the need to change the statement. We agree that the original phrasing may give the impression that the plasma potential is directly linked to dynamic plasma instabilities, which is not the intent of our statement.

We have removed the respective statement and revised the text in **Section 2.1**.

7. Page 5: The plasma is not generated instantaneously. There is a period of plasma expansion during which the floating potential changes significantly. The description here is misleading and also gives the impression that the floating potential is constant (see for example Fig. 2).

We thank the reviewer for pointing this mistake out. The statement has been modified accordingly as follows: During HiPIMS, the discharge at the magnetron is pulsed, consequently, the substrate is immersed in the sputter plasma predominantly during the sputter pulses. The HiPIMS discharge is initiated as soon as power is applied to the target followed by a period of plasma expansion during which the floating potential U_F also increases, as apparent in **Figure 2**. SFP-HiPIMS makes use of this temporally restricted negative floating potential to accelerate ions when they arrive at the substrate.

8. Page 7: The delta in delta_t is commonly associated with a time interval. The authors use delta_t when referring to a time instant. This might confuse the reader into thinking that different pulse lengths were used.

We thank the reviewer for this valuable suggestion. It is true that Δt is sometimes also used in the context of time intervals. To avoid confusion we changed the definition of this time interval. Consistent with existing literature on MIS-HiPIMS we are now using time offset t_{offset} throughout the manuscript.

"The synchronization condition can be described relative to the start of the main HiPIMS pulse that supplies the film-forming ions which should be accelerated. The goal of SFP-HiPIMS is to accelerate the film-forming ions from this main HiPIMS pulse by the floating potential induced by a subsequent HiPIMS pulse. We define the time offset t_{offset} as the time between the start of these two pulses."

9. Figure 2: I cannot understand why the authors see little to no argon during the AI-HiPIMS pulses (and no difference when delaying the AI-HiPIMS pulse). How was this mass spec measurement carried out?

We thank the reviewer for their comment. We do observe the presence of Ar during the AI-HiPIMS pulses, a portion of which is accelerated by the floating potential generated by the AI pulses. This acceleration is unavoidable to some extent; however, based on the ToF data collected using the mass spectrometer, only a fraction of the ionized flux of Ar ions is accelerated. To maintain clarity and avoid cluttering the graph, we intentionally focused on presenting the relevant ions from the Sc pulse, directing the reader's attention to key information. In response to the reviewer's comment, we have now included the ToF data for 20 μs AI pulses in the supplementary information for completeness (see **Figure S1**).

The mass spectrometry measurements were performed using an in-situ time-resolved mass spectrometer, with details of the setup and procedure provided in the Methods section of the manuscript. Additionally, a more detailed explanation of the measurement methodology can be found in the main text and supplementary information of one of our previous publications, which has been appropriately referenced (<https://doi.org/10.1016/j.surfcoat.2023.129719>).

10. Pages 7-8: It would be of great interest to know the Ar content in the deposited samples, i.e., how it varies vs delay time.

We thank the reviewer for their comment. In the revised version of the manuscript, we provide a detailed analysis of the Ar-content in all films. The SFP-HiPIMS show no measurable Ar content via RBS/ERDA and XPS (**Figure S6** and **Figure S7**). In addition, the film properties are best, when the timing of the floating potential

is optimally matched to the metal-ion arrival at the substrate. These measurements support that preferential metal-ion acceleration is achievable during SFP-HiPIMS. In contrast to Sync. Mode I-IV, considerable amounts of Ar are present in the films deposited using RF substrate bias. This can be explained by the higher Ar ion energy during RF substrate biasing (see **Figure S8**).

Answer to the reviewer comments for NCOMMS-24-52890A

REVIEWER COMMENTS

Reviewer #1 (Remarks to the Author):

The authors made a considerable improvement of their manuscript by including additional measurements with RF bias, remeasuring the floating potential and by addressing all received comments. In my opinion, the results are method oriented and I still disagree with the following statement included in the abstract: "The results of this study demonstrate that SFP-HiPIMS provides a practical and economical solution for a long-standing challenge in physical vapor deposition, which can be implemented in standard deposition equipment." TOF measurements are not easy and one can expect significant deviations when changing the experimental arrangement. HiPIMS operated at low pressures generates already very energetic species able to compact the films. In conclusion I see a certain impact but less moderate than claimed.

Answer:

We thank the reviewer for their positive feedback on the revised version of our manuscript.

Regarding the practicality of our approach: We appreciate this concern, and we do acknowledge that time of flight measurements are not trivial. In addition, they require dedicated equipment which might not be available in every lab. Therefore, we decided to tone down the claims on practicality. These changes are now reflected in the abstract and conclusion:

"The results of this study demonstrate that SFP-HiPIMS provides a solution for a long-standing challenge in physical vapor deposition, which can be implemented in standard deposition equipment."

"In conclusion, SFP-HiPIMS provides a solution for the tailored acceleration of ions onto insulating substrates, an important challenge in ionized physical vapor deposition (IPVD)."

Regarding the comparison of SFP-HiPIMS to conventional HiPIMS at low process pressures: We have performed related experiments in our own lab. While we see a small improvement when performing HiPIMS at lower pressures the results are not comparable to ion acceleration via negative substrate bias potentials.

Reviewer #2 (Remarks to the Author):

The authors have done a good job in responding to the reviewers' critical points, and the additional material is valuable.

Answer:

We thank the reviewer for their positive feedback on our revisions and their valuable suggestions to further improve the manuscript.

While I think the work deserves publication because it contains interesting results, I'm still not convinced that the synchronization claim in its current form is valid. I base this primarily on a point made by me and perhaps even clearer by reviewer 3, namely that the establishment of the floating potential is the result of the sheath physics: the floating potential is the result of the balance of electrons and ions arriving at the surface leading to net zero current, or, as reviewer 3 put it, to reduce the loss of electrons to maintain quasineutrality. Either way, the floating potential does not "linger" for much time as it continuously adjust itself to the plasma and sheath conditions. The wording "The floating potential is mostly constrained to the duration of the HiPIMS pulse and occurs almost immediately when the sputter pulse is applied." is on the one hand trivial because when the plasma is present, the surface potential shifts automatically to a negative value due electrons having a higher mobility than ions and the requirement to maintain net zero current for an insulating surface. The wording is also not quite appropriate because the surface has always a potential. The floating potential (of a surface) does not appear and then disappears. What the authors mean is that the surface potential shifts to negative values in the presence of plasma relative to this plasma. The observation that a negative floating potential appears during the plasma pulse is well known. The correlation of plasma presence and negative floating potential is not sufficiently well shown in Fig. 1 because the sheath in front of the substrate is missing and therefore there is no correct display of the potential distribution.

Answer:

We thank their suggestions on improving the clarity the description of the underlying mechanism related to the plasma sheath formation at the substrate. It is of course true that a plasma sheath forms at an electrically isolated object which is immersed in a plasma. The formation of this sheath is critical as the ion acceleration happens predominantly in this region. This was not appropriately described in the previous version of the manuscript. We have amended the description of the underlying mechanisms in the revised version of the manuscript.

Importantly we have changed Figure 1 to include the plasma sheath at the substrate to give a more correct display of the potential distribution as suggested:

Editorial Note: The figure on this page of the Peer Review File is adapted from Handbook of Deposition Technologies for Films and Coatings, Third Edition, Walten et. al, Chapter 2 – Plasmas in Deposition Processes, Page 50, 2010, with permission from Elsevier²⁰.

In addition, we changed the description of the emergence of the negative floating potential in several places of the manuscript. The changes are highlighted in the manuscript. The most important amendments are listed below:

1. We changed the language in the abstract to make it clear that the floating potential is always there but shifts to negative values during the HiPIMS pulse:

"This negative potential is a result of an imbalance of electron and ion-fluxes to the substrate and is effectively restricted to the duration of each HiPIMS pulse. Here we demonstrate that this transient negative potential can be used to accelerate positively charged ions onto the substrate. By synchronizing the ion arrival with the temporal evolution of the substrate's floating potential, specific ions can be accelerated preferentially, thereby enhancing adatom mobility and improving film quality while reducing the detrimental effects of energetic Ar⁺ ion bombardment."

2. We changed the language in the introduction to describe that a plasma sheath is formed when an electrically isolated object is immersed in a plasma.

A practical approach to induce a negative charge on an electrically isolated surface is to expose it to a plasma. When an isolated object is immersed in a plasma it receives a higher flux of electrons than ions due to the electrons' higher thermal velocity. As a result, a plasma sheath is formed around the substrate and the surface charges negatively. As the negative potential increases, so do the electron repulsion and ion acceleration until eventually a net zero current is reached.

3. We changed the language in the chapter 2 to clarify that the ion acceleration happens predominantly in the sheath region at the substrate.

When an electrically isolated substrate is exposed to a sputter plasma, it acquires a negative floating potential (U_F) due to the higher mobility of electrons compared to ions, leading to the formation of a plasma sheath. Since the floating potential is typically lower than the plasma potential (U_P), the resulting potential difference ($U_P - U_F$) accelerates positive ions across the sheath region towards the substrate surface.²⁰

4. We expanded our description of the temporal evolution of the floating potential during the HiPIMS. This way we don't only state that the negative floating potential is restricted to the duration of the HiPIMS pulse but rather also explain why this phenomenon occurs.

During HiPIMS, the discharge at the magnetron is pulsed; consequently, the substrate is immersed in the sputter plasma predominantly during the sputter pulses. The HiPIMS discharge is initiated as soon as power is applied to the target followed by a period of rapid plasma expansion. As the plasma density increases during the pulse, the electron flux to the substrate also increases, leading to a deepening of the negative floating potential, as apparent in **Figure 2**. As the floating potential becomes more negative, both ion bombardment and electron repulsion intensify, eventually leading to a dynamic equilibrium of the floating potential. When the HiPIMS pulse ends, the plasma density decreases, and the floating potential gradually returns to its pre-pulse state. SFP-HiPIMS makes use of this temporally restricted negative floating potential to accelerate specific ions when they arrive at the substrate.

The effect seen, however, is real, and I believe that a possible explanation, qualitatively in line with the authors, is that the arrival of the metal ions of magnetron 1 is timed when there are rather hot electrons from magnetron 2. The latter gives rise to a negative surface potential and that accelerates whatever ions are present at this time at the substrate sheath edge. The acceleration per is not selective of specific ions but ALL ions present at the sheath edge are accelerated. Since the plasma composition is time-dependent, one may achieve selectivity by timing the plasma composition. This has been shown by timing applied bias pulse in relation the HiPIMS pulse [G. Greczynski, et al., J. Appl. Phys. 121, 171907 (2017)]. Here, an attempt is made to achieve a similar effect for an insulating substrate by using more than one pulsed magnetron. In other words, the authors have made interesting experiments and showed interesting and beneficial effects, the explanation however falls short because the sheath physics above the substrate surface is not properly explained.

Answer:

We thank the reviewer for this valuable comment. We fully agree with the description of the synchronization. Depending on the reference point there are different ways to think about the synchronization. Either the negative floating potential can be timed to match the arrival-time of certain ionic species at the substrate, or the plasma composition in the sheath region of the substrate can be controlled accordingly.

Another important point raised by the reviewer is that the negative floating potential always accelerates ALL ions in the sheath region. This is of course true. The negative substrate bias potential should be timed in a way that only a small amount of thermalized process gas ions is present in the sheath region. Accelerating these low energy ions does not negatively impact the film quality. It is however important to avoid further acceleration of the high energy process gas ions which arrive at the substrate shortly after each HiPIMS pulse. These concepts have been described previously in the context of conventional metal-ion synchronized HiPIMS. While we don't cite the reference provided, we do cite several other similar works by the group of Greczynski who is one of the pioneers in this field. In addition, we reference our own recent works where we demonstrate preferential metal-ion acceleration during HiPIMS of AlN and AlScN. It is important to note that in all of these studies conductive substrates were used.

We changed the language in chapter 2 on page 7 to clarify these points:

"The negative floating potential accelerates all positive ions in the sheath region of the substrate. This also includes the thermalized process gas ions which might still be present during the HiPIMS off-cycle. However, it is possible to time the plasma composition and negative floating potential in a way that leads to the preferential acceleration of certain ionic species. For preferential acceleration of metal-ions, the sputter pulses must be synchronized in a way that a negative floating potential is induced shortly before or in the time frame when the metal-ions are in the direct vicinity of the substrate. "

[...]

"In addition, the synchronization can be adjusted to minimize the acceleration of non-film forming species, such as Ar⁺ or other process gas ions. To this end, especially further acceleration of the energetic process gas ions immediately following the HiPIMS pulse should be avoided.⁸ "

Another remaining point of criticism is related to RF bias versus RF discharge. ("...achieving stable ignition of an RF plasma for substrate biasing"). When plasma is provided, e.g. by the HiPIMS discharge, there is no need to also ignite an RF discharge to get an effect of RF bias. RF bias is supposed to affect the surface potential of the substrate, not to generate ions in an RF discharge. I think the concept of (pulsed) RF bias and (pulsed) RF discharges are mixed up in the description.

Both points of criticism are related because they indicate that the physics of the substrate sheath is not considered as needed.

Answer:

We thank the reviewer for this valuable comment, though we don't agree with it completely. In contrast to DC sputtering the plasma density at the substrate varies significantly during HiPIMS. Especially when working with smaller magnetrons and higher working distances the plasma density at the substrate between HiPIMS pulses is very low. This can be seen well when observing the floating potential over a HiPIMS pulse cycle (see **Figure 2**). The metal-ions, which we are trying to accelerate, arrive predominantly during the off-cycle between two

HiPIMS pulses. In this time interval, even during sputtering from an unbalanced magnetron, the plasma density at the substrate from the magnetron is not sufficient for achieving stable RF biasing. For this reason, an additional RF plasma at the substrate is necessary. In order to ignite this plasma, we can either temporarily increase the power and process gas pressure or provide ions from an external source (e.g. the magnetron discharge). The regions in which a stable RF plasma is achieved are demonstrated in **Fig S4**.

However, we agree, that the sentence "achieving stable ignition of an RF plasma for substrate biasing" is not ideally worded and we have therefore changed it to improve clarity:

"Typically, achieving a stable RF plasma for substrate biasing requires higher pressures and/or power levels, which poses a significant challenge for pulsed RF biasing at standard sputtering pressures (3–5 μ bar)."

Reviewer #3 (Remarks to the Author):

I highly appreciate the efforts the authors have taken in revising the manuscript. Their work, particularly in carrying out additional experiments on floating potential and RF biasing, adds significant value to the study (see also comments below). The manuscript is now much more complete, and I commend them for their thorough revisions.

Additionally, the authors have addressed all the questions I raised in the previous round, and I am satisfied with their responses. Based on the improvements made, I recommend the manuscript for publication. The authors may consider the following minor comments:

Answer:

We thank the reviewer for this very positive feedback and thank them for their valuable feedback which was instrumental in improving the manuscript.

1. Floating potential: The authors have taken great care to revise their introduction on floating potentials. I highly appreciate their added efforts and new measurements of the floating potential in their HiPIMS discharges.

Page 4, top: Change to 'electrically isolated substrate' instead of only substrate.

Answer:

Thank you for this comment, the change has been performed as requested.

Regarding my previous question on tuning the ion energy using their strategy, the authors suggest that a wider range of the floating potential could be achieved by adjusting the balancing of the magnetron or modifying the deposition geometry. While these methods may indeed work, I am not aware of any significant changes in the floating potential resulting from such adjustments (as also shown in the supplementary material). However, the authors demonstrate that very high bias values are not necessary (not even desired) for the current material system, so this concern is less relevant in this context and can be considered outside the scope of the present study.

2. RF biasing: The authors have done an excellent job revising the introduction on RF substrate biasing. Additionally, they have successfully carried out further experiments to compare their approach (SFP-HiPIMS) with conventional HiPIMS + RF substrate biasing, which adds valuable insight to the work. It is very exciting to see that their SFP-HiPIMS process shows a clear advantage.

Page 4, middle: Another issue of using RF biasing in HiPIMS is the difficulty of maintaining a constant self-bias, which ultimately will affect the ion acceleration. In my opinion, this is even more problematic than generating a low-density substrate argon plasma.

Answer:

This is a very good point. While we have been able to achieve relatively stable self-bias values with the experimental setup reported in the study, we also tried other RF power supplies which produced a range of different self-bias potentials over the HiPIMS pulse cycle. We therefore fully agree that even achieving a stable RF bias during HiPIMS is not trivial.

As suggested by the reviewer we have added this point as an additional challenge during RF biasing:

"In addition, it is often difficult to maintain a constant DC self-bias, which also affects ion acceleration.²²"

Page 9: How were the RF self-bias values calculated in Figure S4? From Figure S5 it is seen that the self-bias amplitude changes in time. Did the authors pick a time-averaged value?

Answer:

The reported values are time-averaged over the entire HiPIMS cycle. We added a corresponding statement in the figure caption for clarity.

Page 11, bottom-half: Thank you for also measuring the Ar⁺ ion energies when using RF biasing. This is a really valuable contribution. I would, however, add to the explanation on page 11 that the higher Ar⁺ ion energies observed during RF biasing is likely due to the high accelerating voltages seen on the oscilloscope viewgraphs of the RF bias (Figure S5), where the self-bias voltage is not constant but rather increasing throughout the entire pulsed RF bias period. It is therefore very likely that a fraction of ions will be accelerated to higher energies than desired.

Answer:

Thank you for this important point. We have amended the manuscript on page 11 accordingly:

"This high value can be explained by the high acceleration voltages during RF biasing and the variations in self-bias over the HiPIMS cycle (see **Figure S5**)."